# Solar and Wind Energy Forecasting for Green and Intelligent Migration of Traditional Energy Sources

Syed Muhammad Mohsin [1,2,*](image), Tahir Maqsood [3](image) and Sajjad Ahmed Madani [1]

1 Department of Computer Science, COMSATS University Islamabad, Islamabad 45550, Pakistan
2 College of Intellectual Novitiates (COIN), Virtual University of Pakistan, Lahore 55150, Pakistan
3 Department of Computer Science, COMSATS University Islamabad, Abbottabad 22060, Pakistan
* Correspondence: syedmmohsin9@yahoo.com

**Abstract:** Fossil-fuel-based power generation leads to higher energy costs and environmental impacts. Solar and wind energy are abundant important renewable energy sources (RES) that make the largest contribution to replacing fossil-fuel-based energy consumption. However, the uncertain solar radiation and highly fluctuating weather parameters of solar and wind energy require an accurate and reliable forecasting mechanism for effective and efficient load management, cost reduction, green environment, and grid stability. From the existing literature, artificial neural networks (ANN) are a better means for prediction, but the ANN-based renewable energy forecasting techniques lose prediction accuracy due to the high uncertainty of input data and random determination of initial weights among different layers of ANN. Therefore, the objective of this study is to develop a harmony search algorithm (HSA)-optimized ANN model for reliable and accurate prediction of solar and wind energy. In this study, we combined ANN with HSA and provided ANN feedback for its weights adjustment to HSA, instead of ANN. Then, the HSA optimized weights were assigned to the edges of ANN instead of random weights, and this completes the training of ANN. Extensive simulations were carried out and our proposed HSA-optimized ANN model for solar irradiation forecast achieved the values of MSE = 0.04754, MAE = 0.18546, MAPE = 0.32430%, and RMSE = 0.21805, whereas our proposed HSA-optimized ANN model for wind speed prediction achieved the values of MSE = 0.30944, MAE = 0.47172, MAPE = 0.12896%, and RMSE = 0.55627. Simulation results prove the supremacy of our proposed HSA-optimized ANN models compared to state-of-the-art solar and wind energy forecasting techniques.

**Keywords:** renewable energy; forecasting; machine learning; energy efficiency; sustainability; low carbon emission





## 1. Introduction

Increased energy consumption leads to higher fossil fuel consumption. Brown energy is produced using expensive and environmentally damaging fossil fuels including coal, natural gas, and oil. On the other hand, green energy is produced by inexpensive and widely available renewable energy sources (RESs), such as solar and wind energy. When compared to brown energy sources, RESs have a substantially lower carbon emission rate (CER) [1]. In this context, governments and the scientific community face major difficulties related to lowering electricity costs and ensuring environmental sustainability. Due to the long-term effects of carbon emissions from traditional power plants, some countries have imposed large taxes on carbon emissions [2–4]. Therefore, the research community has made tremendous research efforts to reduce electricity costs and carbon emissions [5–8].

State-of-the-art literature recommends integration of RESs with brown energy sources [9–19] to meet users' energy demand in an environmental friendly and cost efficient manner. Solar and wind energy are the abundantly available main sources of renewable energy. However, both sources are inherently highly variable due to volatile weather

conditions. Figures 1 and 2 show how intermittent energy is generated from solar and wind energy sources. Weather data are taken from the measurement and instrumentation data centre at the national renewable energy laboratory [20], as these data are sufficiently accurate [21]. The intermittency of solar and wind generation creates significant difficulties in seamless integration of solar and wind power into the existing power system [22].

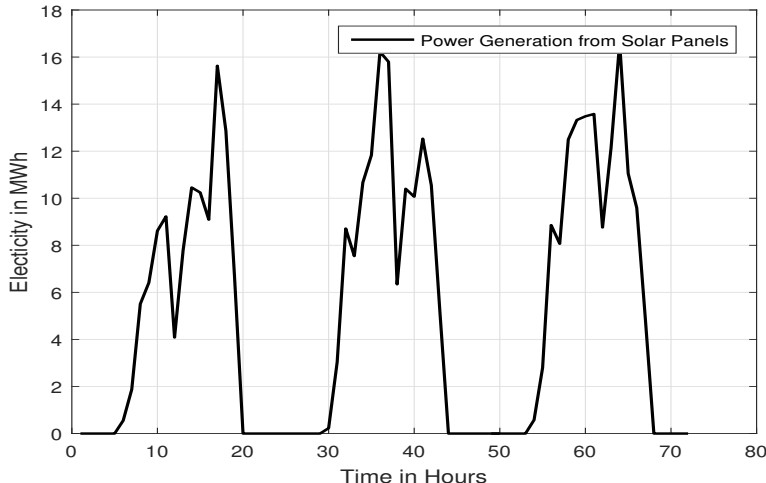

**Figure 1.** Pattern of solar power generation.

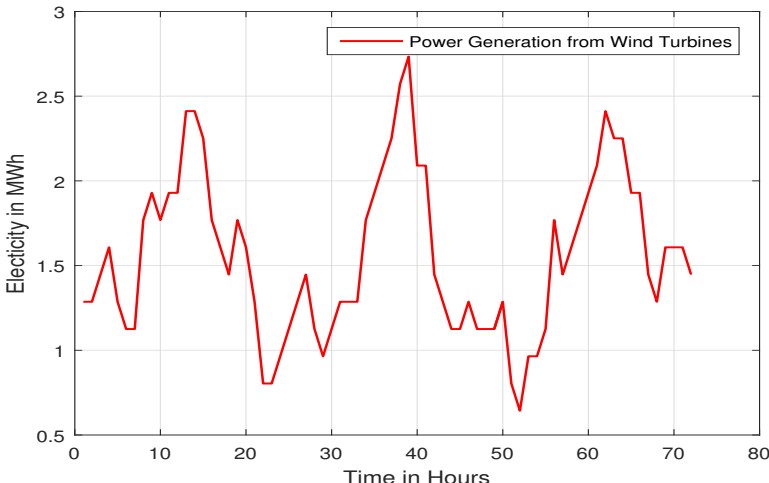

**Figure 2.** Pattern of wind power generation.

Meteorological conditions affect the green energy generation, and green energy production is different in various time zones [23–27]. Therefore, green energy produced by RESs is intermittent in nature. That is a big challenge, especially, while utilizing RESs as the only power source. Consequently, integration of RESs with brown energy is mostly studied and recommended in the literature to cope up with the intermittent nature of RESs. As the renewable energy generation is dependent upon wind direction, wind speed, solar radiance, temperature, humidity, and other weather conditions, it is unreliable [28]. Hence, accurate forecasting of energy production by RESs is necessary to:

1. Minimize the carbon emission,
2. Decrease operational cost of the grid,
3. Trustworthy and safe operations of the power grid
4. Minimize the gap between electricity demand and supply,
5. Reduce the use of electricity reserves through improved scheduling of generation.

Given the importance of renewable energy forecasting, DeepMind, a subsidiary of Alphabet and Google, has chosen machine learning (ML) for 36-h wind energy forecasts to ensure the availability of clean and carbon-free wind energy [29]. Machine learning, which encompasses a variety of other areas, such as data mining, image and speech recognition, optimization, virtual personal assistants, fraud detection, product recommendations, self-driving cars, and artificial intelligence, can be used to process extensive historical Big Data to solve data-driven problems [30,31]. During training, machine learning approaches search for relations between input and output data and make predictions based on the input data. Model generalisation and feature extraction are two areas where machine learning outperforms traditional statistical predictive models.

Machine learning can help make smarter, faster, data-driven estimates about how electricity generation can meet electricity demand [32]. ML can be used for a variety of energy-related tasks, including demand-side management, energy theft detection, demand forecasting, energy generation forecasting, energy price forecasting, predictive maintenance and control, prediction of weather phenomena and optimised energy storage operation that could impact energy forecasting and build energy management. All forms of renewable energy, including hydro, marine, wind, solar, geothermal, bio, hydrogen, and hybrid, can be harnessed with AI models [33].

In the literature, a number of ML forecasting methods for renewable energy have been put forth, and many patents have been registered in this regard. The inventors of US patent US 2015/0186904 A1 at [34] have invented a system for managing and predicting solar and wind energy. They have proposed a current–voltage curve of a solar cell, a diagram to illustrate energy management and use of energy generated by renewable energy sources. US patent 2005/0039787 A1 [35] presents a tool to help grid operators plan and allocate generation resources in a power grid with solar generation capacity on an hourly basis and a week in advance. Tools are also provided to help entities involved in the generation, distribution, and sale of electric energy.

Another invention, patent WO 2011/124226 A1 at [36], discloses a forecasting technique for wind power generation. The invention discusses establishment of a forecasting site at a given location and collecting power generation data from a series of wind turbines, with the first wind turbine at a first location and the second wind turbine at a second location. The estimation is based on the performance data of the group of wind turbines extrapolated into the future or used in conjunction with the location of the forecast site.

Artificial neural networks (ANN) and time series methods, such as autoregressive integrated moving average (ARIMA), are among the most popular ML-based forecasting techniques [37]. The authors of [38] found that time series techniques such as ARIMA lose accuracy when dealing with noisy data and are less accurate than ML techniques. However, ANN may also lose its prediction accuracy due to the high uncertainty of the input data and the random determination of the initial weights between different layers [38].

In this study, we consider reliable and accurate forecasting of solar and wind energy. Previously, the weights to the edges of ANNs were randomly assigned for solar and wind energy forecasting, while in this study we use a meta-heuristic optimization algorithm called harmony search algorithm (HSA) [39] for optimal weights assignment to the edges of ANN for improved forecasting. The main contributions of this study are given below.

- We summarize the state-of-the-art literature on solar and wind energy forecasting.
- During the literature review, we find out that artificial neural networks lose prediction accuracy when dealing with highly intermittent data, such as solar irradiance and wind speed.
- We propose a reliable solar irradiance and wind speed forecasting algorithm named HSA-optimized ANN.
- In our proposed forecasting model, we use HSA for assignment of optimized weights to the edges of ANN.

The rest of the article is organized as follows. State-of-the-art literature review is presented in Section 2. Motivation and problem statement are described in Section 3, and

Section 4 provides a thorough description of an artificial neural network, the harmony search algorithm, and our proposed system model. The simulation setup and methodology for this study are covered in Section 5, while Section 6 presents the simulation findings and pertinent discussions. The study is finally concluded along with future research directions in Section 7.

## 2. Literature Review

Prominent machine learning based solar and wind energy forecasting models [40–48] and [49–58] are briefly explained in Sections 2.1 and 2.2, respectively. The summary of the literature review on solar and wind energy forecasting studies is given in Tables 1 and 2, respectively.

### 2.1. State-of-the-Art Literature on Solar Energy Forecasting

An efficient and effective building energy management system (EMS) can be developed with a reliable energy supply system. Photovoltaic (PV) generation is intermittent; hence, its reliable and accurate forecasting is very important in the development of an efficient EMS. Authors of [40] have proposed a probabilistic day-ahead PV generation forecasting model. A clear sky model is transformed into temperature and shading, and then its deviation is used to train a bagging regression tree for point forecasting of PV energy. A proposed probabilistic forecast model was tested for one year in Munich, Germany, and results proved its accuracy in point forecasting for energy management system applications.

The paper [41] developed and evaluated a daily global solar radiation model from the European centre for medium range weather forecasting by using an ANN-based machine learning model. They compared the ANN model with other models, namely support vector regression, genetic programming, and gaussian process machine learning. Mean absolute error (MAE) and root mean square error (RMSE) were implemented for benchmarking. Results concluded that the ANN-based prediction model was better than other data-driven prediction models.

Solar irradiance is affected by meteorological factors, such as temperature, humidity, cloud cover, dust in desert locations, and sunshine intensity. As a result, solar output varies. Authors of [42] used aerosol optical depth and angstrom data for an hour-ahead solar irradiance forecasting. The proposed forecasting model was compared with different data-driven forecasting models, namely k nearest neighbors, multilayer perception, and support vector regression model. The proposed model was tested on Saudi Arabia data, and it was concluded that it is superior to compared forecasting models, especially for desert areas.

Photovoltaic cells produce electric power when exposed to sun rays. The relationship between energy supply and demand needs to be optimized by reliable solar energy forecasting. The authors of [43] proposed a multi-variant neural network ensemble framework trained on meteorological data. After combining the results with Bayesian model averaging, the proposed technique was compared with real-time solar PV data from the University of Queensland. Validation of the proposed framework was performed by one-day ahead forecasting. Results prove that the proposed multi-variant neural network ensemble framework helps improve the accuracy of PV power output.

Hourly solar irradiation for the following day was predicted by the authors of [44] using LSTM. Inputs include data from the weather forecast for the following day, which includes information on temperature, humidity, sky coverage, wind speed, and precipitation. The model was trained with data from multiple locations of Korea Meteorological Administration. It was found that the proposed model had strong forecasting capabilities with RMSE of 30 $W/m^2$.

**Table 1.** Summary of literature on solar energy forecasting.

| Paper | Energy Source | RES Forecasting | Implementation Strategy | Objective(s) | Dataset Type | Performance/Result |
|-------|---------------|-----------------|-------------------------|--------------|--------------|--------------------|
| [40] | Solar energy | Yes | Probabilistic day ahead PV forecast | PV energy forecasting | Historical | Continuous ranked probability skill score = 90.94% |
| [41] | Solar energy | Yes | Global solar radiance prediction | Daily global solar radiance | Historical | ANN RMSE = 1.613, ANN MAE = 1.146 |
| [42] | Solar energy | Yes | Aerosol optical depth (AOD) and angstrom data for solar irradiance forecasting | One hour solar irradiance prediction | Historical | MLP RMSE = 32.75 ($W/m^2$) |
| [43] | Solar energy | Yes | Multivariate neural network ensemble framework | PV output power forecast | Historical | MAPE = 3.1 |
| [44] | Solar energy | Yes | LSTM model for solar irradiance forecasting | Accurate forecast of solar irradiance | Historical | RMSE = 30 $W/m^2$ |
| [45] | Solar energy | Yes | LSTM model for solar irradiance forecasting | Forecasting of solar irradiance | Historical | 3.2% improvement in nRMSE over the SVR model |
| [46] | Solar energy | Yes | FFNN and LSTM | Accurate forecast of solar irradiance | Historical | Combination of MM and MO performed better |
| [47] | Solar energy | Yes | Image-based dataset and LSTM model | Solar irradiance forecasting | Historical | Pearson Product-Moment Correlation Coefficient (PCCs) is used in this study |
| [48] | Solar energy | Yes | GRU, LSTM, RNN, SVR, and FFNN | Accurate forecast of solar irradiance | Historical | GRU is better than LSTM |

Using an LSTM model, the authors of [45] predicted hourly solar radiation for the city of Johannesburg. Solar radiation, temperature, daylight hours and relative humidity were used as training inputs for the LSTM network. Model was build using solar radiation data of National Oceanic and Atmospheric Administration from 2009 to 2019. The simulation results showed that the proposed LSTM network had a 3.2% improvement in normalised RMSE over the SVR model.

To anticipate solar radiation over a multilevel horizon in northern Italy, the authors of [46] used two different neural network types, FFNN and LSTM. The proposed models used a variety of methods, including multi-model (MM) and multi-output (MO), to build their predictive models. Six years of weather data from 2014 to 2019 was collected from the Italian weather station used in this study. Historical solar radiation data were used to train the model. Comparative results of the study proved that the proposed models performed better by combining the techniques of MM and MO.

The authors of [47] proposed a new method of solar irradiance prediction using an image-based dataset and LSTM model. The developed model can predict solar radiation up to 60 min in advance. The LSTM model was introduced with two different methods based on the input variables. Prediction results of the second model were better. Authors of [48] used the historical data from Korea Department of Meteorological Administration SURFRA system to analyze different deep learning and machine learning solar irradiance

prediction algorithms. Simulation results showed that performance of deep learning models was better.

### 2.2. State-of-the-Art Literature on Wind Energy Forecasting

Due to its wide availability and limitless supply, wind energy is a particularly popular source of energy. Production of wind energy is hampered by uncertainty of air flow/pressure, among other things. The authors of [49] performed probabilistic wind speed forecasting through an ensemble model. The proposed ensemble model is composed of a recurrent neural network, wavelet threshold de-noising (WTD), and an adaptive neural fuzzy inference system (ANFIS). Sub-model variance is used to calculate wind speed forecasting, which is then confirmed for one hour wind speed prediction. Accuracy of the proposed model over its counterparts was proved by simulation results.

Wind energy is highly dependent on wind speed, wind direction, weather temperature, and weather pressure that make it unpredictable, hence unreliable. In [50], the authors exploited ANN to measure different local meteorological conditions that affect wind flow. In order to predict the wind speed, MAE and RMSE were determined. Reliable wind speed forecasting is needed to plan, develop, and monitor an intelligent power system. As the wind energy relies upon wind speed, pressure, temperature, and wind direction, its forecasting mechanism was proposed by the authors of [51]. Raw data is decomposed using an empirical wavelet transformation in a deep-learning-based hybrid wind speed forecasting model. The proposed model was validated in a way that simulation results show highly accurate wind speed prediction.

The authors of [52] stated that wind energy generation, conversion, and optimal control are dependent on reliable wind speed prediction. They proposed EnsembleLSTM using non-linear learning to predict the wind speed. Long short-term memory (LSTM) neural network neurons and numerous hidden layers are used in the suggested method to help reliable wind speed prediction. Later on, the wind speed forecasting process involves the usage of support vector regression machines and external optimization methods. The proposed method was compared with two cases of Inner Mongolia, China, for 10 min ahead and one hour ahead forecasting. Results proved the efficacy of the proposed method.

Wind energy has economical and environmental advantages, so it has garnered much attention of policy makers and the research community. However, uncertainty in wind power generation is unacceptable and a challenging task to overcome. A deep-learning-based ensemble solution was proposed by the authors of [53] for probabilistic wind power forecasting. In order to deal with uncertainties, this study proposed an enhanced point forecasting technique based on wavelet processing and convolutional neural networks (CNN) for wind energy forecasting. The non-linearity of each frequency also increased predicting accuracy. Results show that the suggested technique outperforms its competitors.

Wind power generation is economical and environment friendly; however, irregular wind power generation leads to peak load pressure and frequency regulation issues at grid stations. Wind power forecasting can make its supply steady. Therefore, for accurate wind power prediction, the authors of [54] suggested a long short-term memory improved forget gate network model. Results demonstrate significant improvement in prediction accuracy and speed up in the convergence process.

The authors of [55] presented the SSA-EMD-CNNSVM model, which uses singular spectrum analysis (SSA) for noise reduction and trend extraction from actual data. Time empirical mode decomposition (EMD), as the name suggests, is used to separate time series of wind speed into sublayers. Following that, a convolutional support vector machine (CSVM) is used to forecast wind speed. The proposed prediction model was compared with other wind speed prediction models, including the CNNSVM, EMD-BP, SVM, and EMD-Elman models. Results demonstrated the superiority of the proposed model.

Table 2. Summary of literature on wind energy forecasting.

| Paper | Energy Source | RES Forecasting | Implementation Strategy | Objective(s) | Dataset Type | Performance/Result |
|---|---|---|---|---|---|---|
| [49] | Wind energy | Yes | WTD–RNN–ANFIS based ensemble model | Reliable wind speed forecasting | Historical | MAE ANN = 0.929, MAE SVM = 0.963, RMSE ANN = 1.293, RMSE SVM = 1.349 |
| [50] | Wind energy | Yes | Wind speed forecasting through ANN | Forecasting of wind speed | Historical | RMSE = 0.675, MAE = 0.536 |
| [51] | Wind energy | Yes | Wind speed prediction through wavelet transformation and recurrent neural networks | Wind speed is predicted | Historical | Wind speed series 1, MAPE ARIMA = 7.17, MAE ARIMS = 0.93, RMSE ARIMA = 1.21 |
| [52] | Wind energy | Yes | Ensemble LSTM using non-linear learning to predict the wind energy | Wind speed forecasting | Historical | MAE EnsemL-STM = 0.574, RMSE EnsemL-STM = 0.755, MAPE EnsemLSTM = 5.41 |
| [53] | Wind energy | Yes | Deep learning based ensemble approach for probabilistic wind power forecasting | Wind power forecasting | Historical | Performance improvemney by 48.42%, 45.02%, and 45.10% as compared to three benchmarks |
| [54] | Wind energy | Yes | Long short-term memory enhanced forget gate network model for reliable wind power prediction | Wind power forecasting | Historical | 18.3% rise in accuracy |
| [55] | Wind energy | Yes | Convolutional support vector machine (CNNSVM) | Wind speed forecasting | Historical | RMSE = 39.25%, MAE = 39.21% , MAPE = 42.85% |
| [56] | Wind energy | Yes | SVM-based prediction and MLP are used | Wind speed forecasting | Historical | MSE SVM = 0.78%, MSE MLP = 0.9% |
| [57] | Wind energy | Yes | Wavelet transform | Wind speed forecasting | Historical | MAPE increased from 14.79% to 22.64% |
| [58] | Wind energy | Yes | ARIMA and ANN | Wind speed forecasting | Historical | MAPE = 6.97% |

The support vector machine method is used in [56] for wind prediction. The regression analysis is performed after mapping the time series data for any variable into a higher dimensional space (e.g., Hilbert space), according to the procedure. In addition, the results of SVM-based prediction and multilayer perceptron (MLP) models were compared. MSE and RMSE were the performance measures used in [56]. Since the SVM had a mean square error of 0.78% compared to the MLP of 0.9%, it was found that the SVM performed better than the MLP.

Using a wavelet transform, the authors of [57] deconstructed a wind series. The selection of input parameters for the SVM was supported by the genetic algorithm approach. The input must be improved to select the best forecast candidates. According to the results,

persistence increased MAPE from 14.79% to 22.64%, while WT-SVM-GA did not. The NNWT technique, in which the prediction for the next three hours is made using the historical data of the last twelve hours, was compared with ARIMA (1,2,1) and NN by [58]. The MAPE value was found to be 6.97% using the NNWT technique.

The authors of [59] discussed the benefits and limitations of solar energy in detail. A brief description along with benefits and limitations of solar energy and wind energy are briefly described in Table 3.

**Table 3.** Short description and merits/limitations of solar energy and wind energy.

| Type | Short Description | Benefits | Limitations |
|---|---|---|---|
| Solar energy | Energy of sunrays is transformed into electricity with the help of PV cells | • Inexhaustible energy source<br>• Pollution-free energy<br>• Directly exploitable and widely available<br>• Renewable energy<br>• Being labor intensive industry, improves job opportunities<br>• Reduces electricity cost | • Huge initial installation cost<br>• Dependent over climate and weather<br>• Intermittent in nature<br>• Performance issues of batteries and inverters, etc.<br>• Shortage of skilled manpower |
| Wind energy | Produced by kinetic energy caused by flow of air on the surface of Earth | • Clean energy<br>• Carbon free generation<br>• Minimizes dependence over fossil fuels | • Intermittent in nature<br>• Dependent over air dynamics such air flow, pressure, direction, humidity etc. |

## 3. Motivation and the Problem Statement

### 3.1. Motivation

Many researchers have considered the energy optimization and environmental implications caused by excessive brown energy usage. A few have proposed using both energy sources, while some have recommended using RESs only. Recent research articles cited at [40–58] have focused on solar energy and wind energy forecasting, respectively. The literature review motivated us towards accurate and reliable RES forecasting because it is very important for effective and efficient grid management. Moreover, it is helpful in minimizing user energy cost, reducing carbon emissions, overcoming energy imbalances, decreasing dependence upon electricity reserves, and better scheduling of different energy sources.

### 3.2. Problem Statement

Accurate renewable energy forecasting is important for the minimization of user energy cost and carbon emission. ANN-based renewable energy forecasting techniques lose prediction accuracy due to uncertainty of input data and random determination of initial weights between different layers of the ANN. Therefore, the objective of this work is to develop "a harmony search algorithm optimized artificial neural network model for reliable and accurate solar and wind energy forecasting".

## 4. Proposed System Model

### 4.1. Artificial Neural Network

A collection of linked nodes referred to as artificial neurons makes up an artificial neural network, which functions similarly to the human nervous system. Artificial neural nodes are connected to each other through edges, and the edges carry signal or output to the next artificial neuron where some logical action is performed [37]. This signal is possibly sent to the next neuron for further processing or final output. ANN may have a single hidden layer or multiple hidden layers, and each layer may have a different number of nodes. The quantity of hidden layers, learning rate, and iterations are the main governing factors of an ANN. The activation functions that have an impact on ANN processing include softmax, sigmoid, gaussian error linear units, exponential linear units and swish, among others. Figure 3 depicts the basic architecture of ANN.

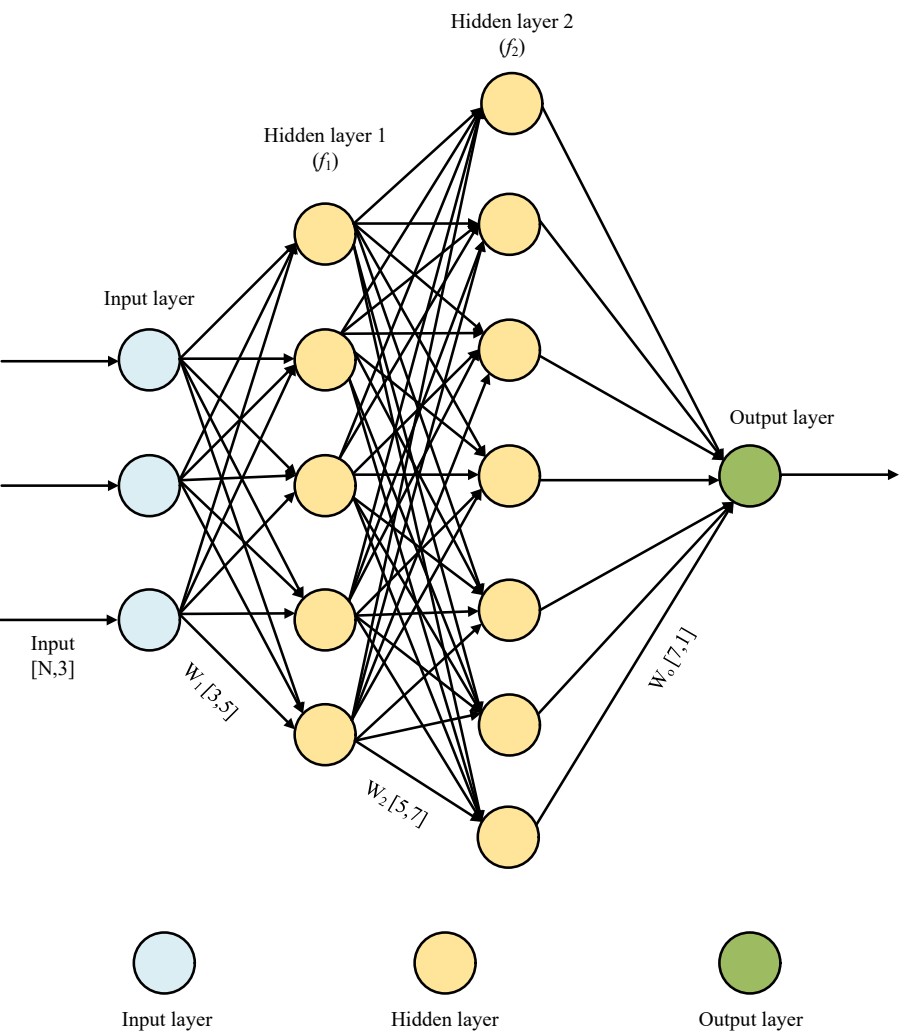

**Figure 3.** Basic architecture of artificial neural network.

### 4.2. Harmony Search Algorithm

Harmony search algorithm (HSA) is a nature inspired evolutionary meta-heuristic optimization algorithm [39], proposed by Zong Wo Geem et al. [60] in 2001. Harmony improvisation is the term used to describe the process by which artists apply this algorithm to enhance their harmony. Every time a musical band where each performer plays a different instrument completes the harmony improvisation process. Each member of the musical ensemble serves as a decision variable in this situation, and each musical instrument has a different pitch. Successful musical harmony is achieved throughout the process of improvising harmony, and this successful harmony is then updated in the harmony memory (HM). HM contains the top solution vectors. The HSA process is illustrated in Figure 4, and Table 4 displays manually chosen HSA control parameters in the context of Equations (1)–(5). According to Table 4, *HMS*, *NI*, *HMCR*, *PAR*, $PAP_{max}$, and $PAI_{max}$ stand for harmony memory size, number of iterations, harmony memory consideration rate, pitch adjustment rate, maximum pitch adjustment proportion (used for continuous variables), and maximum pitch adjustment index (used for discrete variables), respectively.

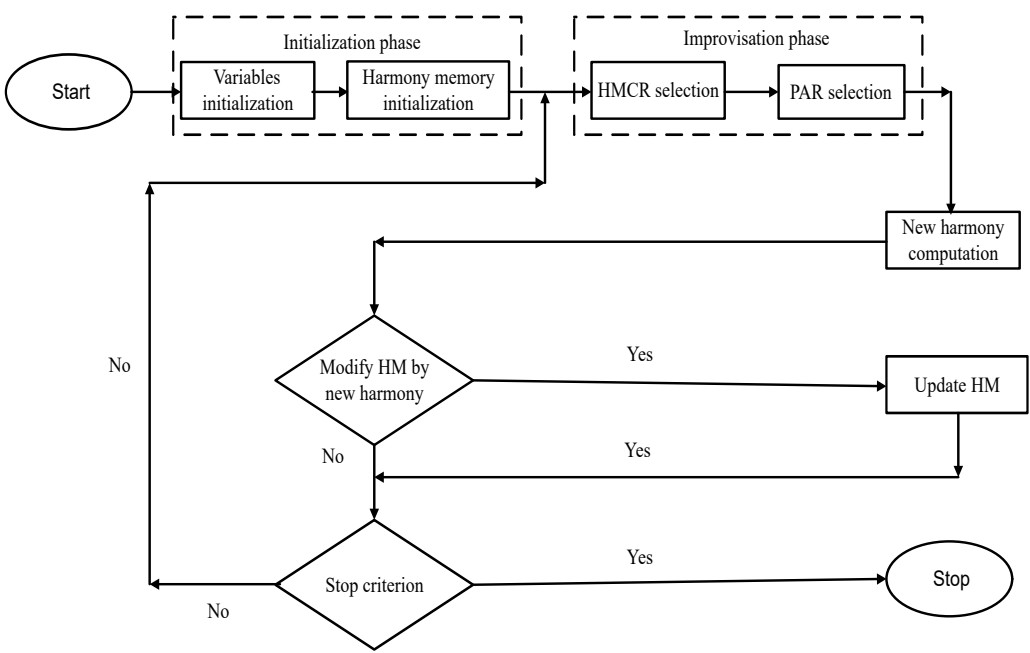

**Figure 4.** Procedural steps of harmony search algorithm.

**Table 4.** HSA control parameters.

| Control Parameter | Value |
|---|---|
| *HMS* | 3 |
| *NI* | 10 |
| *HMCR* | 0.9 |
| *PAR* | 0.5 |
| $PAP_{max}$ | 0.25 |
| $PAI_{max}$ | 2 |

A brief description of each procedural step of HSA is given in the following.

1. Variables initialization
   Values limit of different variables used in HSA are defined during this step.

   - Upper and lower limit of variables

   $$x_i^L \leq x_i \leq x_i^U \tag{1}$$

   - Harmony memory size (*HMS*)

   $$10 \leq HMS \leq 3 \tag{2}$$

   - Harmony memory consideration rate (*HMCR*)

   $$0.0 \leq HMCR \leq 1.0 \tag{3}$$

   - Pitch adjustment rate (*PAR*)

   $$0.0 \leq PAR \leq 1.0 \tag{4}$$

   - Maximum number of iterations (*NI*), i.e., stopping criteria

   $$0 \leq NI \leq 10 \tag{5}$$

2. HM initialization
   The harmony memory matrix is randomly initialized during the HM initialization step using following equation.

$$x_{(i,j)} = l_j + rand().U_j - l_j \tag{6}$$

   The *j*th element of the initial harmony memory in Equation (6) is indicated by $x_{(i,j)}$, whereas the rand() function is used to generate random values between zero and one. $U_j$ and $l_j$ in Equation (6) represent upper and lower bounds of variables, respectively.

3. *HMCR* selection
   A random number between zero and one is created during the *HMCR* selection phase of the HSA improvisation phase using the rand() function, as indicated in Equation (7). The value for that specific place is chosen if the randomly generated value is smaller than the *HMCR*; otherwise, a new random number is generated.

$$V_{i,j} = \begin{cases} x(randj) & if\ randb()\ is < HMCR \\ l_j + rand().U_j - l_j & else \end{cases} \tag{7}$$

4. *PAR* selection
   *PAR* selection is another sub-part of the the HSA improvisation phase in which the memory elements selected during the *HMCR* step are further improved in the *PAR* selection step. The *PAR* selection step works on the basis of Equation (8); *bw* in Equation (8) represents bandwidth which plays an important role in pitch adjustment.

$$V_{i,j} = \begin{cases} V_i^j rand().bwj & if\ rand()\ is < PAR \\ V_i^j & else \end{cases} \tag{8}$$

5. Update HM
   Upon successful completion of the new harmony selection process, it is updated in the HM by replacing the already present worst memory there.

6. Checking the stop criteria
   The harmony improvisation process terminates at the maximum number of iterations (*NI*), i.e., stopping criteria, as shown in Equation (5).

The nature-inspired meta-heuristic algorithm HSA offers perfect stability between the search process's exploration and exploitation stages [61]. Moreover, HSA has been effectively used in a variety of application areas, including image processing, wireless sensor networks, text clustering, and fuzzy clustering [62]. Therefore, it has high precision, faster convergence speed, and less complexity. Consequently, we selected HSA for weight optimization of the ANN edges in this study.

### 4.3. Proposed System Model

Recent research has focused on the integration of brown energy sources and RESs (solar and wind) because of low operational cost and carbon-free production of RESs. However, reliable forecasting of RESs is an important issue and needs keen attention. We have considered an ANN-based solar and wind energy forecasting model for efficient solar and wind energy production, thereby efficient supply and demand management, less energy cost, and less carbon emissions.

The literature review revealed that machine learning techniques are much better than time series techniques for solar and wind forecasting. The authors of [38] stated that machine learning techniques, such as, artificial neural network suffer from:

1. Loose precision due to high uncertainty of input data like solar and wind energy production;
2. Random determination of initial weights between different layers can affect the performance of an ANN.

In this study, we propose HSA-optimized ANN for solar and wind energy forecasting models, where initial weights between different layers of ANN are determined by a meta-heuristic algorithm named the harmony search algorithm. Our proposed forecasting model has strengths of machine learning (ANN) and meta-heuristic algorithm (HSA). Consequently, our proposed forecasting algorithm has high precision, faster convergence speed, and less complexity. Our proposed forecasting model for reliable solar and wind energy prediction is shown in Figure 5.

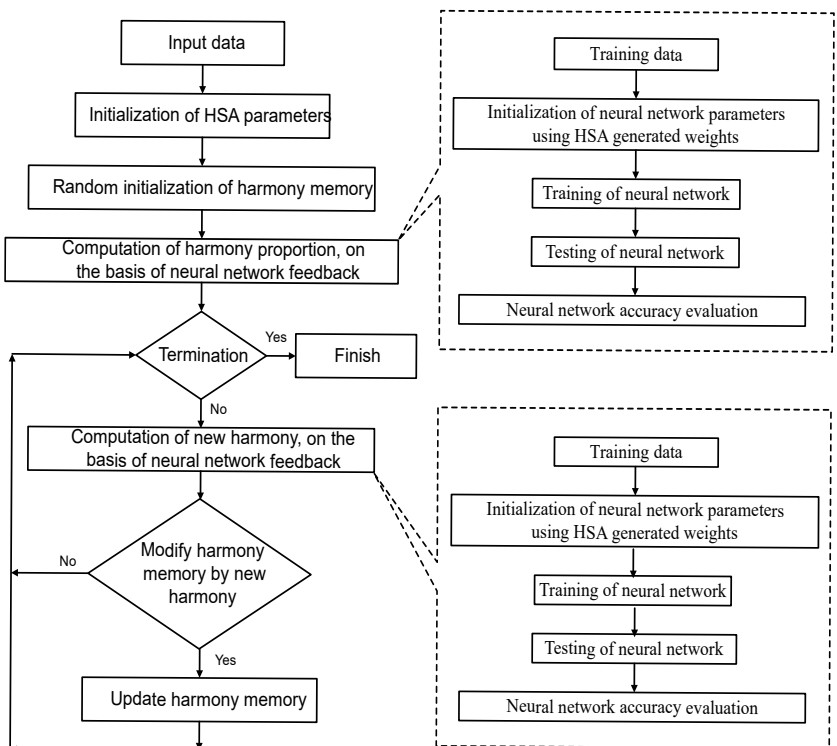

**Figure 5.** Proposed solar and wind energy forecasting model.

In Figure 5, it is evident that weights at the edges of ANN are being adjusted by the meta-heuristic algorithm HSA which is very famous for optimization problems. Here, the optimal weights assignment to the ANN's edges has a favourable impact on the forecasting of solar and wind energy. The findings mentioned in Section 6 contain supporting data.

## 5. Simulation Setup and Methodology

### 5.1. Simulation Setup

In this section, implementation specifications of our suggested model are described in terms of their performance indicators. Our model is tested on a system with a core i7, 16 GB of RAM, and a 4.8 GHz processor. Python and the Anaconda IDE environment are employed. Table 5 describes the simulation settings of our proposed system model.

Overfitting occurs when the model curve becomes too complex and performs too well on training data but fails or degrades performance on test data. The main cause of overfitting is that the model has not learned well from the training data. When underfitting occurs, the model does not perform well even on the training data because the model is too simple and/or the input features are not very expressive. If the number of epochs is too high, the model may overfit, and if the number of epochs is very low, the model may underfit. To avoid overfitting, we used an early stopping criterion in our model. If the model does not perform better after a certain number of epochs, e.g., between 50 and 60 epochs, it is automatically stopped even if the fixed number of epochs is 100. We tested our model for different numbers of epochs, i.e., from 100 to 300, and we found 200 to be

the optimal number of epochs where we obtained good results for training and testing of the data.

**Table 5.** Simulation parameters.

| Parameter(s) | Setting |
|---|---|
| Neural network | Artificial neural network |
| Number of hidden layers | 2 |
| Heuristic algorithm | Harmony search algorithm |
| Optimizer | Adam |
| Loss functions | MSE, MAE, MAPE, RMSE |
| Batch size | 4 |
| Number of epoch | 200 |

*5.2. Methodology*

First, solar and wind energy datasets were downloaded from [63] for the time period of 1 January 2015 to 1 March 2018. A total of 70% of the total data was utilized for training purpose whereas, 30% of the data was used for testing the accuracy of the proposed forecasting model. Pre-processing of solar and wind energy datasets was performed using standard scalar to improve the training of our proposed model by means of data standardization. The following four different forecasting models were developed.

1. ANN-based solar irradiance forecasting model without HSA (using random weights assignment at the edges of ANN layers)
2. ANN-based wind speed forecasting model without HSA (using random weights assignment at the edges of ANN layers)
3. ANN-based solar irradiance forecasting model with HSA (using HSA optimized weights assignment at the edges of ANN layers)
4. ANN-based wind speed forecasting model with HSA (using HSA optimized weights assignment at the edges of ANN layers)

A basic structure of the ANN model with 2 hidden layers was created, and solar and wind datasets downloaded from [63] were loaded. The model was trained with 70% of the data ,and performance was measured by means of error criteria, i.e., mean square error (MSE), mean absolute error (MAE), mean absolute percentage error (MAPE) and root mean square error (RMSE). Later on, we used the harmony search algorithm for assignment of optimized weights at the edges of the ANN instead of random weight assignment to the edges of the ANN. Different weights harmonies were created using the harmony search algorithm, and this process was repeated in a loop until the number of iterations, i.e., 5 in our case. During this process, each time a new harmony (weight) was generated it was fitted to the ANN to obtain a loss value. Loss values of all the harmonies (weights) were compared and finally, the best harmony (weight) among all was selected and applied to the edges of the ANN. Tables 6 and 7 show the simulation results.

**Table 6.** Performance evaluation of solar irradiance forecasting.

| Error Criteria | ANN [38] | GA Optimized ANN [38] | ANN [41] | SVR [41] | ANN (Ours) | HSA Optimized ANN (Proposed) |
|---|---|---|---|---|---|---|
| MSE | 0.53 | 0.29 | — | — | 0.06354 | 0.04754 |
| MAE | 0.53 | 0.29 | 1.146 | 1.367 | 0.18520 | 0.18546 |
| MAPE | 7.6% | 4.5% | — | — | 0.32430% | 0.32475% |
| RMSE | 0.62 | 0.37 | 1.613 | 1.994 | 0.25208 | 0.21805 |

**Table 7.** Performance evaluation of wind speed forecasting.

| Error Criteria | ANN [49] | SVM [49] | GRNN [51] | EWT-Elman [51] | ANN (Ours) | HSA Optimized ANN (Proposed) |
|---|---|---|---|---|---|---|
| MSE | — | — | — | — | 0.46358 | 0.30944 |
| MAE | 0.929 | 0.963 | 0.89 | 0.66 | 0.66419 | 0.47172 |
| MAPE | — | — | 6.88% | 5.08% | 0.13988% | 0.12896% |
| RMSE | 1.293 | 1.349 | 1.27 | 0.83 | 0.68087 | 0.55627 |

## 6. Results and Discussions

### 6.1. Solar Irradiance Forecasting

As mentioned above, we proposed and developed an HSA-optimized ANN model for solar irradiance forecasting, and we used well-known error value measurement methods for accurate and reliable analysis of the results. Solar irradiance prediction was carried out for one week of all seasons, i.e., autumn, spring, summer, and winter.

In Table 6, the solar irradiance forecasting accuracy of our proposed model is compared with solar irradiance forecasting accuracy of the study at [38]. The authors of [38] developed two models for solar irradiance forecasting: (1) an ANN model was used with random determination of weights at its edges (ANN [38]) and (2) a genetic algorithm (GA) was used for optimized weights assignment at the edges of ANN (GA optimized ANN [38]). The authors of [41] implemented ANN and SVR models for solar irradiance forecasting. Instead, we used; (1) an ANN model with random determination of weights at its edges (ANN (Ours)) and (2) an ANN with HSA-optimized weight assignment at its edges (HSA-optimized ANN (proposed)). Simulation results prove the supremacy of our proposed solar irradiance forecasting model.

The results reported in the Table 6 show that the solar irradiance prediction accuracy is higher with our proposed HSA-optimized ANN model, achieving MSE = 0.04754, MAE = 0.18546, MAPE = 0.32430%, and RMSE = 0.21805. On the other hand, the first competitor ANN at [38] achieved the result values of MSE = 0.53, MAE = 0.53, MAPE = 7.6%, and RMSE = 0.62. Its second competitor, GA-optimized ANN, at [38] achieved results of MSE = 0.29, MAE = 0.29, MAPE = 4.5%, and RMSE = 0.37. Its third competitor ANN at [41] achieved results of MAE = 1.146 and RMSE = 1.613, whereas it fourth competitor SVR at [41] achieved results of MAE = 1.367 and RMSE = 1.994. Results of all the competitors of our proposed HSA-optimized ANN model are far behind. The authors of [41] did not consider MSE and MAPE as evaluation criteria in their study. During solar irradiance forecasting simulations, the computational time of ANN (Ours) was recorded = 60 s, whereas the computational time of our proposed HSA-optimized ANN was recorded = 176 s. The computational time of the HSA-optimized ANN is higher because it involves another meta-heuristic algorithm (HSA) for optimal weight assignment.

Figure 6 shows the result graph of actual solar irradiance values, ANN-predicted solar irradiance values without HSA, and ANN-predicted solar irradiance values with HSA for the whole dataset. The results of the one-week solar irradiance forecast for autumn, spring, summer, and winter seasons are shown in Figure 7a–d, respectively. The green lines in these figures represent the actual solar irradiance values, and the blue lines show the predicted solar irradiance values using ANN without HSA. The red lines, on the other hand, show the predicted solar irradiance values of the ANN model optimized with HSA, i.e., our proposed model.

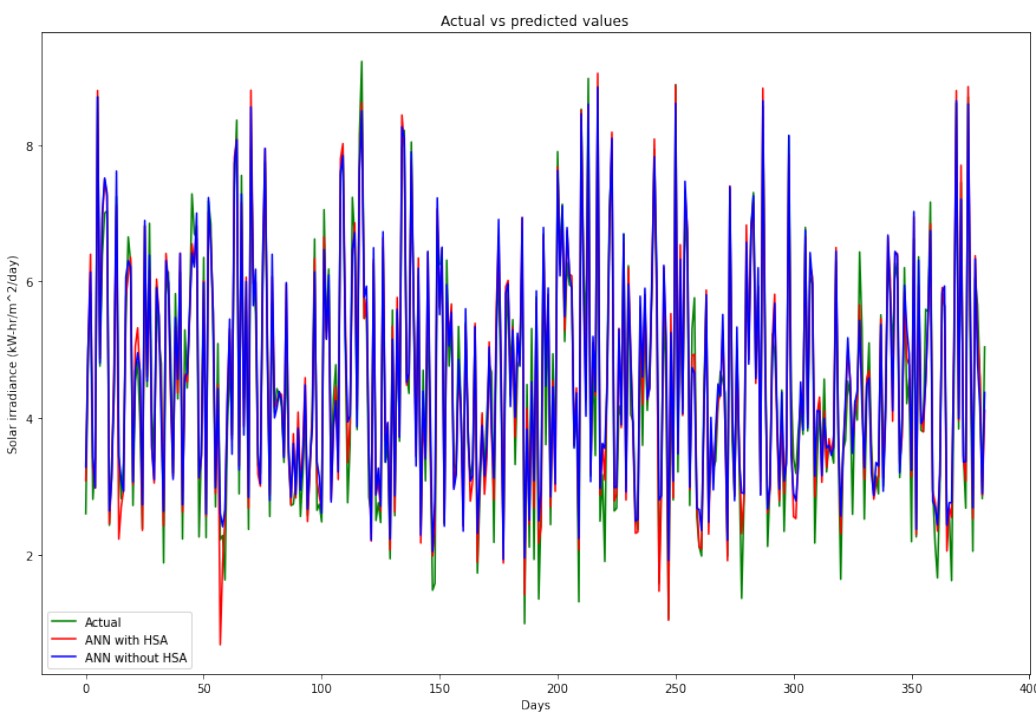

**Figure 6.** Actual vs. forecasted solar irradiation.

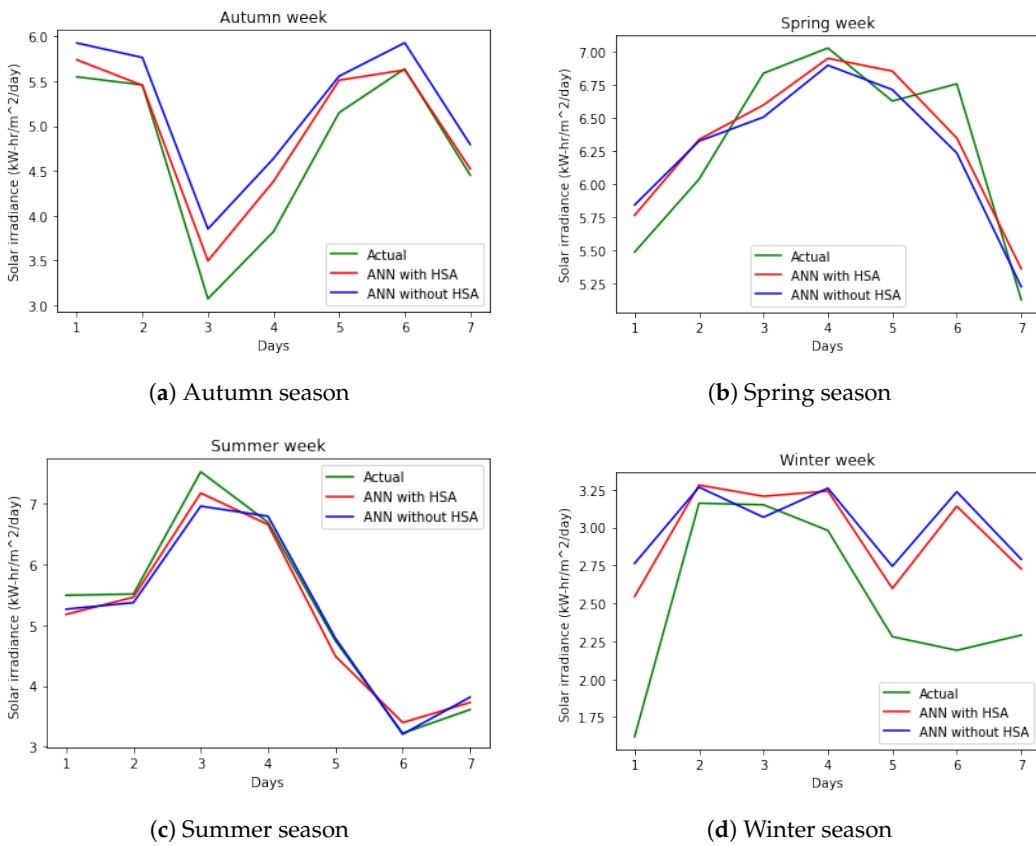

(**a**) Autumn season

(**b**) Spring season

(**c**) Summer season

(**d**) Winter season

**Figure 7.** Actual vs. forecasted 1-week solar irradiation of different seasons.

In Figure 7a–d it can be seen that sometimes the line of actual values (green line) and the line of predicted values (blue and red lines) cross each other. Ideally, this should not have happened. As mentioned earlier, whole dataset is split into 70% for training and 30% for accuracy testing purpose. We tested our models for a one-week solar irradiance forecast

of four seasons. Each forecast model makes its predictions based on its learning process, where it learns about trends in changes. However, actual weather is 100% unpredictable, and we see sudden changes in actual weather parameters such as temperature, solar irradiance, etc. Therefore, the actual and predicted lines overlap in the results. We will deal with this issue in our future work.

*6.2. Wind Speed Forecasting*

Wind energy is less reliable compared to solar energy [22]. Therefore, results accuracy in the case of wind speed forecasting is less compared to solar energy forecasting. Wind speed prediction was carried out for one week of all seasons, i.e., autumn, spring, summer and winter. In Table 7, the wind speed forecasting accuracy of our proposed model is compared with wind speed forecasting accuracy of the studies at [40,51]. The authors of [49] discussed different wind speed forecasting models, and we have selected two representative models which are ANN and SVM. The authors of [51] discussed two models for wind speed forecasting: (1) a GRNN model and (2) an EWT-Elman model. We used: (1) an ANN model with random determination of weights at its edges (ANN (Ours)) and (2) an ANN with HSA-optimized weight assignment at the edges of the ANN (HSA-optimized ANN (proposed)). Simulation results prove the supremacy of our proposed wind speed forecasting model.

The results reported in the Table 7 show that the wind speed prediction accuracy is higher with our proposed HSA-optimized ANN model, achieving MSE = 0.30944, MAE = 0.47172, MAPE = 0.12896%, and RMSE = 0.55627. On the other hand, Its first competitor ANN at [49] achieved the values of MAE = 0.929 and RMSE = 1.29, whereas it second competitor SVM at [49] achieved the values of MAE = 0.963 and RMSE = 1.349. Its third competitor, GRNN, at [51] achieved the values MAE = 0.89, MAPE = 6.88%, and RMSE = 1.27, and the fourth competitor, EWT-Elman, at [51] achieved the values of MAE = 0.66, MAPE = 5.08%, and RMSE = 0.83. Results of all competitors of our proposed HSA-optimized ANN model were far behind. MSE was not considered by the authors of [49,51]. The authors pf [49] also did not consider MAPE in their study. However, we considered MSE as well in our proposed model, and its result values are shown in Table 7. During wind speed forecasting simulations, the computational time of ANN (Ours) was recorded = 60 s, whereas the computational time of our proposed HSA-optimized ANN was recorded = 323 s. Computational time of HSA-optimized ANN was higher because it involved another meta-heuristic algorithm (HSA) for optimal weight assignment.

Figure 8 shows the resuls graph of actual wind speed values, ANN predicted wind speed values without HSA, and ANN predicted wind speed values with HSA for the whole dataset. The results of the one-week wind speed forecast for autumn, spring, summer, and winter seasons are shown in Figure 9a–d, respectively. The green lines in these figures represent the actual wind speed values, and the blue lines show the predicted wind speed values using ANN without HSA. The red lines, on the other hand, show the predicted wind speed values of the ANN model optimized with HSA, i.e., our proposed model.

In Figure 9a–d it can be seen that sometimes the line of actual values (green line) and the line of predicted values (blue and red lines) cross each other. Ideally, this should not have happened. As mentioned earlier, whole dataset is split into 70% for training and 30% for accuracy testing purpose. We tested our models for a one-week wind speed forecast of four seasons. Each forecast model makes its predictions based on its learning process, where it learns about trends in changes. However, actual weather is 100% unpredictable and we see sudden changes in actual weather parameters such as wind speed, wind direction, temperature, wind pressure, etc. Therefore, the actual and predicted lines sometimes overlap each other in the results. However, we will try our best to deal with this problem in our future work.

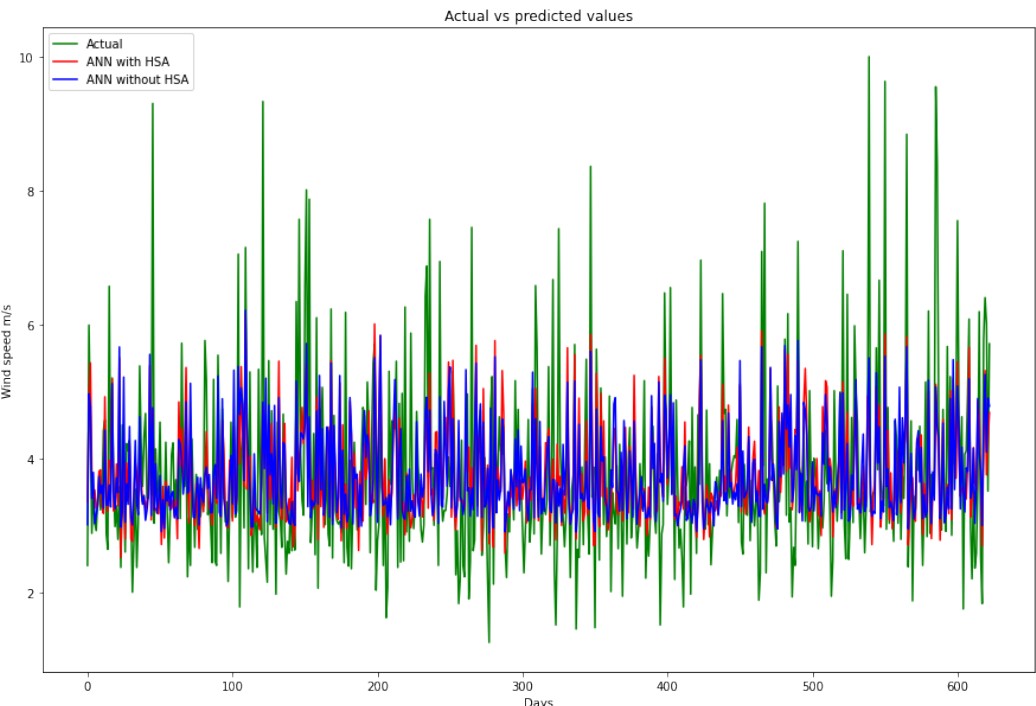

**Figure 8.** Actual vs. forecasted wind speed.

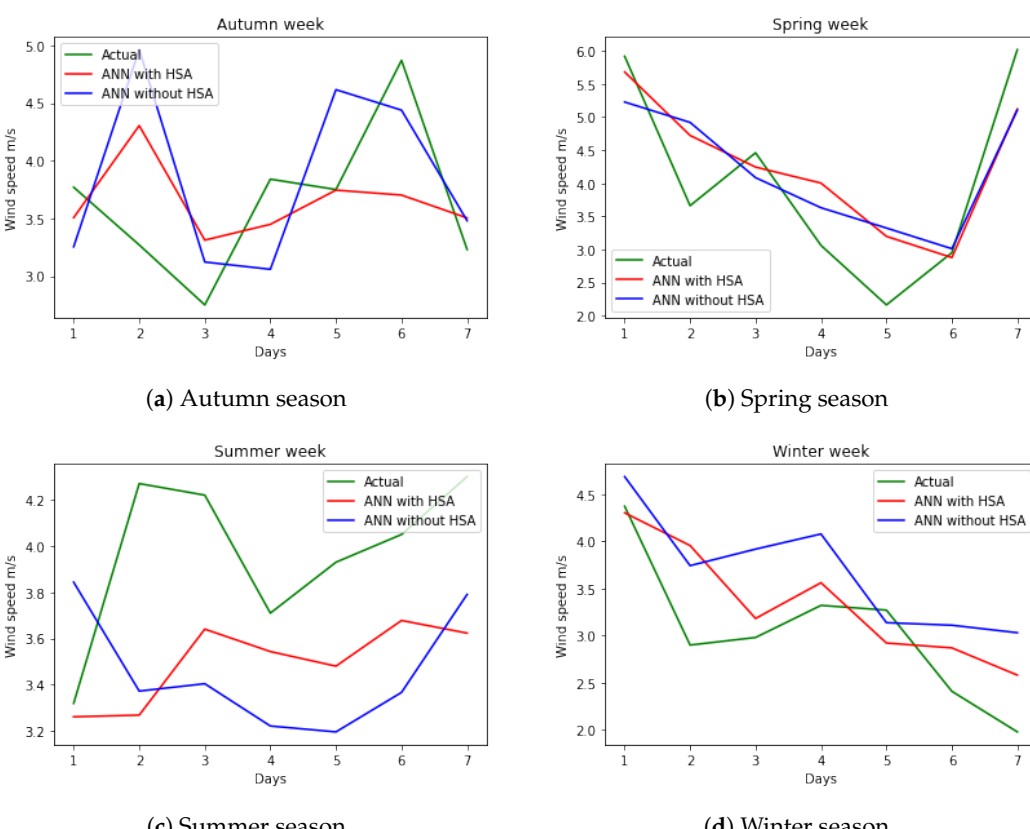

(**a**) Autumn season

(**b**) Spring season

(**c**) Summer season

(**d**) Winter season

**Figure 9.** Actual vs. forecasted 1-week wind speed of different seasons.

## 7. Conclusions and Future Work

Fossil fuel generated electric power leads to higher energy cost and environmental pollution. To deal with higher electricity costs and environmental implications, solar and wind energy are abundantly available renewable energy sources being used for green environment and low cost energy. As solar and wind energy are highly intermittent in

nature, in this study we have proposed HSA-optimized ANN solar irradiance and wind speed forecasting models. We have assigned HSA-optimized weights at the edges of ANN layers, and simulation results prove the accuracy of our proposed forecasting models. Our proposed HSA-optimized ANN model for solar irradiation forecast achieved the values of MSE = 0.04754, MAE = 0.18546, MAPE = 0.32430(%), and RMSE = 0.21805, whereas our proposed HSA-optimized ANN model for wind speed prediction achieved the values of MSE = 0.30944, MAE = 0.47172, MAPE = 0.12896(%), and RMSE = 0.55627. Result accuracy of our proposed wind speed forecasting model is less compared to our proposed solar irradiance forecasting model. Accuracy enhancement of our proposed wind speed forecasting model is our future work. Furthermore, identifying the causes of the volatile character of wind speed and solar irradiance is also essential since doing so enables the adaptation or even mitigation of the intermittent nature of wind and solar energy.

**Author Contributions:** Methodology, S.M.M., T.M. and S.A.M.; Software, S.M.M.; Validation, T.M. and S.A.M.; Formal analysis, S.M.M. and T.M.; Investigation, T.M. and S.A.M.; Data curation, S.M.M.; Writing—original draft, S.M.M.; Writing—review & editing, T.M.; Visualization, S.M.M.; Supervision, T.M. and S.A.M.; Project administration, T.M. and S.A.M.; Funding acquisition, S.M.M. All authors have read and agreed to the published version of the manuscript.

**Funding:** This research received no external funding.

**Data Availability Statement:** Not applicable.

**Conflicts of Interest:** The authors declare no known conflict of interest.

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
