# Peer review of "Solar and Wind Energy Forecasting for Green and Intelligent Migration of Traditional Energy Sources"

_sustainability, doi:10.3390/su142316317_

Round 1

Reviewer 1 Report

Dear Authors please 

The Figure 1 and Figure 2 solar and wind energy graph is for Pakistan or for all other countries. Please it should be addressed well otherwise it confuse to readers

2) Authors why the present study is focused in solar and wind authors should address. Moreover, add a separate sections about solar energy and wind energy. For clear understanding as shown in below

2.1. Solar energy: Solar energy is considered one of the better energy due to viable cheap. Moreover, the disadvantages of solar energy is the rise in ambeint temp the efficiency will be decrease about 0.4-0.5%. Therefore, a lot of cooling systems available in energy markets (Something like this) and for authors convenience a few articles are below which will help and must cite  

a) https://doi.org/10.1016/j.heliyon.2021.e07920

b) https://doi.org/10.3390/inventions6040063

c) Thermal management of solar photovoltaic module to enhance output performance: an experimental passive cooling approach using discontinuous aluminum heat sink

d) Experimental assessment of thermoelectric cooling on the efficiency of PV module

3) Similarly a separate  section for wind energy and few more citations 

4) is their any possibility authors can show figure 6 as fig.6 a,b,c into three different graphs it will take more time for understanding. 

5) Moreover, Fig.6 shift after the statement explanations. it is easy to understand for readers

6) Authors can even try fig.6. fig.7, fig.8 and fig.9 by side side it can reduce the page and easy to compare statement and explantation. A proper discussion should add in the statement, a lot of comparison should be in the statement. why the behavior of curves for each season needed to be well addressed between fig.6. fig.7, fig.8 and fig.9 and missing 

7) Similar comments repears again for wind forcasting in section 6.2 there are lot of discussion needed to be included and addressed but authors were missing in their statement 

8) Please fix all my comments carefully authors 

Author Response

Response to Reviewer 1 Comments

Point 1: The Figure 1 and Figure 2 solar and wind energy graph is for Pakistan or for all other countries. Please it should be addressed well otherwise it confuse to readers.

Response 1: Dear reviewer, we appreciate you reminding us of this overlooked detail. Given that the data from the Measurement and Instrumentation Data Center (MIDC) is sufficiently accurate according to earlier studies [17], we used the meteorological data from the MIDC of the National Renewable Energy Laboratory [16]. Specifically, we used weather data from Sun Spot One, Loyola Marymount University Rotating Shadowband Radiometer, Oak Ridge National Laboratory, and La Ola Lanai, United States between August 1 and August 3, 2018 to represent inherent volatile nature and power generation from solar and wind energy sources in figure 1 and 2, respectively. Relevant references [16] and [17] have been included in the article at appropriate positions.  

Point 2: Authors why the present study is focused in solar and wind authors should address. Moreover, add a separate sections about solar energy and wind energy. For clear understanding as shown in below

2.1. Solar energy: Solar energy is considered one of the better energy due to viable cheap. Moreover, the disadvantages of solar energy is the rise in ambeint temp the efficiency will be decrease about 0.4-0.5%. Therefore, a lot of cooling systems available in energy markets (Something like this) and for authors convenience a few articles are below which will help and must cite  

  1. a) https://doi.org/10.1016/j.heliyon.2021.e07920
  2. b) https://doi.org/10.3390/inventions6040063
  3. c) Thermal management of solar photovoltaic module to enhance output performance: an experimental passive cooling approach using discontinuous aluminum heat sink
  4. d) Experimental assessment of thermoelectric cooling on the efficiency of PV module

Response 2: Thankfully, it is noted that solar and wind energy are abundant, cheap sources of energy, which is why both are the focus of this study. Other renewable energy sources are not the focus of this study. To ensure an appropriate structure for the article, we have included the literature on solar and wind energy forecasts in Section 2 Literature Review. Subsection 2.1 discusses the state of the literature on solar forecasts and subsection 2.2 discusses the state of the literature on wind energy forecasts. In addition, we have gratefully cited the recommended valuable papers at reference numbers [20], [21], [22] and [23].

Point 3: Similarly a separate section for wind energy and few more citations. 

Response 3: Subsection 2.2 discusses the state of the literature on wind energy forecasting. At the recommendation of the reviewer, we have included the following studies (ref. # [53], [54], and [55]) in the wind forecasting subsection.

The Support Vector Machine method is used in [53] for wind prediction. The regression analysis is performed after mapping the time series data for any variable into a higher dimensional space (e.g., Hilbert space), according to the procedure. In addition, the results of SVM-based prediction and multilayer perceptron models (MLP) were compared. MSE and RMSE were the performance measures used in [53]. Since the SVM had a mean square error of 0.78% compared to the MLP of 0.9%, it was found that the SVM performed better than the MLP.

Using a wavelet transform, the authors of [54] deconstruct a wind series. The selection of input parameters for the SVM is supported by the genetic algorithm approach. The input must be improved to select the best forecast candidates. According to the results, persistence increased MAPE from 14.79% to 22.64%, while WT-SVM-GA did not. The NNWT technique, in which the prediction for the next three hours is made using the historical data of the last twelve hours, was compared with ARIMA (1,2,1) and NN by [55]. The MAPE value was found to be 6.97% using the NNWT technique.

Point 4: Is their any possibility authors can show figure 6 as fig.6 a,b,c into three different graphs it will take more time for understanding.

Response 4: Dear reviewer, thank you for your insightful comment. In fact, in Figures 7, 8, 9, and 10, we have shown individual graphs of seasonal solar radiation prediction, while in Figure 6 we have shown the result of combined solar radiation prediction. We hope that you, dear reviewer, will find our rationale acceptable.

Point 5: Moreover, Fig.6 shift after the statement explanations. it is easy to understand for readers.

Response 5: With thanks to the reviewer, Figure 6 has been moved after the explanation for better reader understanding.

Point 6: Authors can even try fig.6. fig.7, fig.8 and fig.9 by side side it can reduce the page and easy to compare statement and explantation. A proper discussion should add in the statement, a lot of comparison should be in the statement. why the behavior of curves for each season needed to be well addressed between fig.6. fig.7, fig.8 and fig.9 and missing. 

Response 6: By appreciating the reviewer feedback regarding the reduction in the number of pages, we have adjusted the figures showing the solar radiation prediction page by page. Further discussion of the behaviour of the curves is mentioned along with Figure 7 for the solar graphs.

Point 7: Similar comments repears again for wind forcasting in section 6.2 there are lot of discussion needed to be included and addressed but authors were missing in their statement. 

Response 7: In appreciation of reviewer feedback regarding the reduction in the number of pages, we have appended figures showing the wind speed predictions side by side. Further discussion of the behaviour of the curves is mentioned along with Figures 9 for the wind diagrams. We hope that the reviewer will find our rationale sufficient and acceptable.

We have tried our best to improve the manuscript and have made recommended changes in the manuscript. These changes do not affect the scope of the paper. The specific position of the changes in the revised article is explained in detail in this red-letter response document.

We sincerely thank the editor and reviewers for their work and hope that the corrections will meet with approval.

Once again, thank you very much for your comments and suggestions.

Reviewer 2 Report

The authors suggest a harmony search algorithm optimized artificial neural networks to predict solar and wind energy. The authors should appropriately revise their manuscript for possible publication.

My specific comments are below:

  1. Please put a key numerical finding in the abstract.
  2. Please briefly explain the benefits of ML techniques to estimate wind and solar energy. You can compare other mathematical and statistical techniques with ML methods.
  3. If you want to increase the impact of your work, you can briefly give a few commercial applications of ML techniques to estimate wind and solar energy. You can also explain a few novel patents about this topic in the first section of the manuscript.
  4. Tables 1 and 2 are nice, but the authors can expand these tables to increase the impact of their study. You can put the input and output parameters on the tables. Also, you can give the performance indicators of the models on the table (numerical values to show the accuracy of different ML techniques).
  5. I suggest the authors expand the literature review in Sections 2.1 and 2.2 because there are many excellent studies about ML techniques usage for wind and solar energy generation in the literature.
  6. Please use some numerical values to support your statement: "ANN-based renewable energy forecasting techniques lose prediction accuracy due to the uncertainty of input data and random determination of initial weights between different layers of ANN."
  7. Please also use the different number of epochs to show the overfitting and underfitting range of your model. Then, you can estimate the optimum number of the epoch. 
  8. Table 6 does not include computational time. At least, you can put the computational time for ANN (yours) and the proposed model (HSA optimized ANN).
  9. Please expand Table 6 and Table 7. There are so many studies about prediction of solar and wind energy by using different ML techniques. 

Author Response

Response to Reviewer 2 Comments

Point 1: Please put a key numerical finding in the abstract.

Response 1: Dear reviewer we are thankful to you. We have included following numerical findings in the abstract to make it more clear and understandable.

Extensive simulations were performed and our proposed HSA-optimized ANN model for solar radiation prediction achieved MSE = 0.04754, MAE = 0.18546, MAPE = 0,32430%, and RMSE = 0.21805, while our proposed HSA-optimized ANN model for wind speed prediction achieved MSE = 0.30944, MAE = 0.47172, MAPE = 0.12896%, and RMSE = 0.55627.

Point 2: Please briefly explain the benefits of ML techniques to estimate wind and solar energy. You can compare other mathematical and statistical techniques with ML methods.

Response 2: With reference to the valuable feedback received from the esteemed reviewer, the following description has been updated along with the relevant references in Section 1 of the revised version.

Machine learning (ML), which encompasses a variety of other areas such as data mining, image and speech recognition, optimization, virtual personal assistants, fraud detection, product recommendations, self-driving cars, and artificial intelligence, can be used to process vast amounts of historical big data to address data-driven challenges. Machine learning approaches aim to identify relationships between input and output data. After training with the training dataset, the prediction input data can be fed into well-trained machine learning models, which can then be used to generate predictions [25] ,[26].

In the literature, a number of ML forecasting methods for renewable energy have been put forth. ML-based methods like artificial neural networks (ANN) and time series methods like autoregressive integrated moving average (ARIMA) are among the most popular forecasting techniques [30]. The authors of [31] found that time series techniques such as ARIMA lose accuracy when dealing with noisy data and are less accurate than ML techniques. However, ANN may also loose its prediction accuracy due to the high uncertainty of the input data and the random determination of the initial weights between different layers [31].

Point 3: If you want to increase the impact of your work, you can briefly give a few commercial applications of ML techniques to estimate wind and solar energy. You can also explain a few novel patents about this topic in the first section of the manuscript.

Response 3: We greatly appreciate your comment as it motivated us to apply for the patent. The commercial application of the techniques of ML has already been briefly discussed (please see response #2). However, the following discussion on the inclusion of some patents (please see [27], [28], and [29]) is included in Section 1 of the revised manuscript.

The inventors of US patent US 2015O186904A1 at [27] have invented a system for managing and predicting solar and wind energy. They have proposed a current-voltage curve of a solar cell, a diagram to illustrate energy management and use of energy generated by renewable energy sources. Patent US 20050O397.87A1 [28] presents a tool to help grid operators plan and allocate generation resources in a power grid with solar generation capacity on an hourly basis and a week in advance. Tools are also provided to help entities involved in the generation, distribution, and sale of electric energy manage their contractual delivery obligations in the hourly day-ahead wholesale market and the spot market in a grid with solar generation capacity.

Another invention, patent WO 2011/124226 A1 at [29], discloses a forecasting technique for wind power generation. A prediction site is placed at a particular geographic location, and the method includes collecting performance data indicative of power generation from a group of wind turbines, a first wind turbine being located at a first site and a second wind turbine being located at a second site. By extending the performance data into the future or using the location of the prediction site, the estimation is based on the performance data from the collection of wind turbines. A method for scheduling wind power generation for a power grid and a wind power generation prediction system that can predict the output power of a wind turbine are also disclosed.

Point 4: Tables 1 and 2 are nice, but the authors can expand these tables to increase the impact of their study. You can put the input and output parameters on the tables. Also, you can give the performance indicators of the models on the table (numerical values to show the accuracy of different ML techniques).

Response 4: We are grateful for this good comment. Certainly we want to increase the significance of our study, so we have included the performance parameters of the cited solar and wind forecasting studies in the corresponding Tables 1 and 2.

Point 5: I suggest the authors expand the literature review in Sections 2.1 and 2.2 because there are many excellent studies about ML techniques usage for wind and solar energy generation in the literature.

Response 5: In the light of reviewer feedback following studies are included in sub-sections 2.1 and 2.2.

Hourly solar irradiation for the following day was predicted by the authors of [41] using LSTM. A number of variables are used as inputs, including information from the next day's weather forecast on temperature, humidity, sky coverage, wind speed, and precipitation. The weather forecast data is provided by the Korea Meteorological Administration. Data from a number of locations within the target area are used to train the model. It was found that the proposed model has strong forecasting capabilities with RMSE of 30 W/{m^2}.

Using an LSTM model, authors of [42] predicted hourly solar radiation for the city of Johannesburg. Temperature, daylight hours, relative humidity, and solar radiation are used as training inputs for the LSTM network. The dataset needed to build the model was collected by the National Oceanic and Atmospheric Administration (NOAA) for a 10-year period, between 2009 and 2019. (NOAA). The simulation results show that the proposed LSTM network has a 3.2% improvement in normalised root mean square error (nRMSE) over the SVR model.

To anticipate solar radiation over a multilevel horizon in northern Italy, the authors of [43] used two different neural network types, FFNN and LSTM. The proposed models use a variety of methods, including multi-model (MM) and multi-output (MO), to build their predictive models. The data used in the study were collected from an Italian weather station at the Como campus for a period of six years, from 2014 to 2019. Historical solar radiation data were used to train the model. The report also includes a comparison of the proposed model with a clear sky and the two persistent models. The simulation results show that the proposed models perform better by combining the techniques of MM and MO.

Using an image-based dataset and an LSTM model, the authors of [42] proposed a new method to predict solar radiation. The developed model can predict solar radiation up to 60 minutes in advance. The LSTM model is introduced with two different methods based on the input variables. The first approach considers the mean, solar irradiance and solar irradiance 5 minutes ahead. The second approach considers the solar irradiance from five minutes ago, the recent solar irradiance, the mean value, the variance value, the red-blue comparison method, and the three-step search method as input parameters. The LSTM model trained with the second method has shown better prediction results.

Authors of [44] have proposed a new method of solar irradiance prediction using an image-based dataset and LSTM model. The developed model can predict solar radiation up to 60 minutes in advance. The LSTM model is introduced with two different methods based on the input variables. Prediction results of the second model were better. Authors of [45] have analysed different deep learning and machine learning algorithms for solar radiation prediction. The model was trained with historical GHI and irradiance data collected by the Korea Department of Meteorological Administration (KDMA) SURFRAD system. Simulation results showed that performance of deep learning models is better.

The Support Vector Machine method is used in [53] for wind prediction. The regression analysis is performed after mapping the time series data for any variable into a higher dimensional space (e.g., Hilbert space), according to the procedure. In addition, the results of SVM-based prediction and multilayer perceptron models (MLP) were compared. MSE and RMSE were the performance measures used in [53]. Since the SVM had a mean square error of 0.78% compared to the MLP of 0.9%, it was found that the SVM performed better than the MLP.

Using a wavelet transform, the authors of [54] deconstruct a wind series. The selection of input parameters for the SVM is supported by the genetic algorithm approach. The input must be improved to select the best forecast candidates. According to the results, persistence increased MAPE from 14.79% to 22.64%, while WT-SVM-GA did not. The NNWT technique, in which the prediction for the next three hours is made using the historical data of the last twelve hours, was compared with ARIMA (1,2,1) and NN by [53]. The MAPE value was found to be 6.97% using the NNWT technique.

Point 6: Please use some numerical values to support your statement: "ANN-based renewable energy forecasting techniques lose prediction accuracy due to the uncertainty of input data and random determination of initial weights between different layers of ANN."

Response 6: Respected reviewer, solar and wind energy, as we all know, are very sporadic in nature. Both exhibit a high degree of volatility, are unpredictable, and range from 0 to 100. Any forecasting model can run into problems when training and testing the data when such volatile input data is present. The accuracy of predictions made by artificial neural networks (ANNs), despite their high reliability, is ultimately compromised by the problems encountered in testing and training the data. Since each model has its own structure, data set, etc., its numerical values may change from case to case. However, the accuracy between our proposed HSA-optimised ANN and the simple ANN can be seen from the results (Pleas see tables 6 and 7).

Point 7: Please also use the different number of epochs to show the overfitting and underfitting range of your model. Then, you can estimate the optimum number of the epoch. 

Response 7: Overfitting occurs when the model curve becomes too complex and performs too well on training data, but fails or degrades performance on test data. The main cause of overfitting is that the model has not learned well from the training data. When underfitting occurs, the model does not perform well even on the training data because the model is too simple and/or the input features are not much expressive. If the number of epochs is too high, the model may overfit, and if the number of epochs is very low, the model may underfit. To avoid overfitting, we used an early stopping criterion in our model. If the model does not perform better after a certain number of epochs, e.g., between 50 and 60 epochs, it is automatically stopped even if the fixed number of epochs is 100. We tested our model for different numbers of epochs, i.e. from 100 to 300, and we found 200 to be the optimal number of epochs where we got good results for training and testing the data.

Point 8: Table 6 does not include computational time. At least, you can put the computational time for ANN (yours) and the proposed model (HSA optimized ANN).

Response 8: We appreciate your valuable feedback. By agreeing with the comment, we have included the following details regrading computational time of solar and wind forecast models in relevant sub-sections i.e. 6.1 for solar irradiance forecasting and sub-section 6.2 for wind speed forecasting.

During solar irradiance forecasting simulations, the computational time of ANN (Ours) was recorded = 60 seconds whereas the computational time of our proposed HSA optimized ANN was recorded = 176 seconds. Computational time of HSA optimized ANN is higher because it involves another meta-heuristic algorithm (HSA) for optimal weight assignment.

During wind speed forecasting simulations, the computational time of ANN (Ours) was recorded = 60 seconds whereas the computational time of our proposed HSA optimized ANN was recorded = 323 seconds. Computational time of HSA optimized ANN is higher because it involves another meta-heuristic algorithm (HSA) for optimal weight assignment.

Point 9: Please expand Table 6 and Table 7. There are so many studies about prediction of solar and wind energy by using different ML techniques. 

Response 9: By thanking dear reviewer, we have expanded table 6 and 7 by icluding the results of study at [38] and [46], respectively. We hope, dear reviewer finds this inclusion sufficienct and acceptable.  

We have tried our best to improve the manuscript and have made recommended changes in the manuscript. These changes do not affect the scope of the paper. The specific position of the changes in the revised article is explained in detail in this blue-letter response document.

We sincerely thank the editor and reviewers for their work and hope that the corrections will meet with approval.

Once again, thank you very much for your comments and suggestions.

Reviewer 3 Report

Dear Authors, The article is interesting, but a few changes need to be made:

1. The authors the topic is up-to-date, but the abstract should be clarified.

2. It is necessary to describe in more detail what research gap is filled by the article and indicate why the research focuses on solar and wind energy.

3. Please describe what your article brings to the topic compared to other published material?

4. Authors should consider extending the methodology and propose a diagram illustrating this aspect.

5. Please explain the advantages and disadvantages of ML techniques for wind and solar energy.

6. Please complete the conclusions so that they are in line with the arguments presented.

7. The literature should be expanded with research carried out in specific areas in selected countries to indicate the development of the renewable energy market, including wind and solar energy. Articles that may be helpful in the analysis of the research areas may be helpful, for example: Analysis and Evaluation of the Photovoltaic Market in Poland and the Baltic States. Energies 2022, 15, 669. https://doi.org/10.3390/en15020669 Wind Energy Market in Poland in the Background of the Baltic Sea Bordering Countries in the Era of the COVID-19 Pandemic. Energies 2022, 15, 2470. https://doi.org/10.3390/en15072470

8. I ask the authors to carefully correct all my comments.

Author Response

Response to Reviewer 3 Comments

Point 1: The authors the topic is up-to-date, but the abstract should be clarified.

Response 1: Dear reviewer, thank you for your encouraging comment. Abstract has been clarified by adding the following numerical data in blue. We hope that the dear reviewer will find this sufficient and acceptable.

Extensive simulations were performed and our proposed HSA-optimized ANN model for solar radiation prediction achieved MSE = 0.04754, MAE = 0.18546, MAPE = 0,32430%, and RMSE = 0.21805, while our proposed HSA-optimized ANN model for wind speed prediction achieved MSE = 0.30944, MAE = 0.47172, MAPE = 0.12896%, and RMSE = 0.55627.

Point 2:  It is necessary to describe in more detail what research gap is filled by the article and indicate why the research focuses on solar and wind energy.

Response 2: Thank you dear reviewer. We really missed to explicitly mention the research contribution. We have included in section 1 the following main contributions of our study.

  • We have summarized state-of-the-art literature on solar and wind energy forecasting.
  • During literature review, we find out that artificial neural network lose prediction accuracy, when dealing with highly intermittent data such as solar irradiance and wind speed.
  • We proposed a reliable solar irradiance and wind speed forecasting algorithm named as HSA optimized ANN.
  • In our proposed forecasting model, we have used HSA for assignment of optimized weights to the edges of ANN.

Among all renewable nergy sources, solar and wind energy has following merits. Please refer to table 3.

  • Inexhaustible energy source
  • Pollution free energy
  • Directly exploitable and widely available
  • Renewable energy
  • Being labor intensive industry, improves job opportunities
  • Reduces electricity cost
  • Clean energy
  • Carbon free generation
  • Minimizes dependence over fossil fuels

Point 3: Please describe what your article brings to the topic compared to other published material?

Response 3: Thank you for your time and feedback. We have used MSE, MAE, MAPE and RMSE (error criteria) to evaluate the performance of our proposed model. We have included the comparison of our proposed prediction model with other published material in Tables 6 and 7. Please see the updated tables in the revised manuscript.

Point 4: Authors should consider extending the methodology and propose a diagram illustrating this aspect.

Response 4: With thanks to the dear reviewer, we have updated the methodology section as follows. We hope it is acceptable to the dear reviewer. For further discussion of the operation and diagram of the proposed system model, see subsection 4.3.

A basic structure of the ANN model with 2 hidden layers is created and solar and wind datasets downloaded from [58] are loaded. Model is trained with 70% of the data and  performance is measured by means of error criteria i.e. methods such as mean absolute error (MAE), mean absolute percentage error (MAPE), mean square error (MSE), and root mean square error (RMSE). Later on, we used harmony search algorithm for assignment of optimized weights at the edges of ANN, instead of random weight assignment to the edges of ANN. Different weights harmonies are created using harmony search algorithm and this process is repeated in loop until number of iterations i.e. 5 in our case. During this process, each time a new harmony (weight) is generated which is fitted to ANN to get a loss value. Loss values of all the harmonies (weights) are compared and finally, the best harmony (weight) among all is selected and applied to the edges of ANN. Simulation results are shown in tables 6 and 7.

 Point 5: Please explain the advantages and disadvantages of ML techniques for wind and solar energy.

Response 5: With reference to the valuable feedback received from the esteemed reviewer, the following description has been updated along with the relevant references in blue color in Section 1 of the revised version. Further discussion of the techniques of ML can be found in subsections 2.1 and 2.2.

Machine learning (ML), which encompasses a variety of other areas such as data mining, image and speech recognition, optimization, virtual personal assistants, fraud detection, product recommendations, self-driving cars, and artificial intelligence, can be used to process vast amounts of historical big data to address data-driven challenges. Machine learning approaches aim to identify relationships between input and output data. After training with the training dataset, the prediction input data can be fed into well-trained machine learning models, which can then be used to generate predictions [25] and [26].

In the literature, a number of ML forecasting methods for renewable energy have been put forth. ML-based methods like artificial neural networks (ANN) and time series methods like autoregressive integrated moving average (ARIMA) are among the most popular forecasting techniques [30]. The authors of [31] found that time series techniques such as ARIMA lose accuracy when dealing with noisy data and are less accurate than ML techniques. However, ANN may also loose its prediction accuracy due to the high uncertainty of the input data and the random determination of the initial weights between different layers [31].

Point 6: Please complete the conclusions so that they are in line with the arguments presented.

Response 6: In the light of worthy reviewer’s comment, we have updated conclusion with following numerical data. We hope, dear reviewer is satisfied with our response and action. 

Our proposed HSA optimized ANN model for solar irradiation forecast achieved the values of MSE = 0.04754, MAE = 0.18546, MAPE = 0.32430%, and RMSE = 0.21805 whereas, our proposed HSA optimized ANN model for wind speed prediction achieved the values of MSE = 0.30944, MAE = 0.47172, MAPE = 0.12896%, and RMSE = 0.55627.

Point 7: The literature should be expanded with research carried out in specific areas in selected countries to indicate the development of the renewable energy market, including wind and solar energy. Articles that may be helpful in the analysis of the research areas may be helpful, for example: Analysis and Evaluation of the Photovoltaic Market in Poland and the Baltic States. Energies 2022, 15, 669. https://doi.org/10.3390/en15020669 Wind Energy Market in Poland in the Background of the Baltic Sea Bordering Countries in the Era of the COVID-19 Pandemic. Energies 2022, 15, 2470. https://doi.org/10.3390/en15072470

Response 7: We appreciate the reviewer feedback. Suggested studies are inline with our work therefore, we have happily included the recommended studies at ref. # [5] and [6].

We have tried our best to improve the manuscript and have made recommended changes in the manuscript. These changes do not affect the scope of the paper. The specific position of the changes in the revised article is explained in detail in this orange-letter response document.

We sincerely thank the editor and reviewers for their work and hope that the corrections will meet with approval.

Once again, thank you very much for your comments and suggestions.

Round 2

Reviewer 1 Report

Accept in present form

Author Response

Response to Reviewer 1 Comments

Point 1: Accept in present form.

Response 1: Dear reviewer, we appreciate you review comments in round 1 that really helped us in article improvement. We are thankful for your favorable feedback in round 2.

Reviewer 2 Report

The authors significantly improved the quality of this version. I have only a minor comment. The explanations about the benefits of ML in the energy field in the introduction are not clear. You can use the references below to explain the benefits of ML better:

Li, J., Herdem, M. S., Nathwani, J., & Wen, J. Z. (2022). Methods and Applications for Artificial Intelligence, Big Data, Internet-of-Things, and Blockchain in Smart Energy Management. Energy and AI, 100208. 

https://www.evwind.es/2019/02/28/google-deepmind-target-machine-learning-at-boosting-wind-energy-value/66291

https://www.azocleantech.com/article.aspx?ArticleID=1518

Author Response

Response to Reviewer 2 Comments

Point 1: The authors significantly improved the quality of this version. I have only a minor comment. The explanations about the benefits of ML in the energy field in the introduction are not clear. You can use the references below to explain the benefits of ML better:

Li, J., Herdem, M. S., Nathwani, J., & Wen, J. Z. (2022). Methods and Applications for Artificial Intelligence, Big Data, Internet-of-Things, and Blockchain in Smart Energy Management. Energy and AI, 100208. 

https://www.evwind.es/2019/02/28/google-deepmind-target-machine-learning-at-boosting-wind-energy-value/66291

https://www.azocleantech.com/article.aspx?ArticleID=1518

Response 1: Dear reviewer, we are very grateful to you. The suggested literature has helped us a lot to realise the benefits of machine learning in the energy field. We have included the following description in the Introduction section along with the suggested references at [28], [31], and [32]. We hope that you, dear reviewer, will find the updated description sufficient and acceptable.

Given the importance of renewable energy forecasting problem, DeepMind, a subsidiary of Alphabet and Google, has chosen machine learning (ML) for 36-hour wind energy forecasts to ensure the availability of clean and carbon-free wind energy [30]. Machine learning, which encompasses a variety of other areas such as data mining, image and speech recognition, optimization, virtual personal assistants, fraud detection, product recommendations, self-driving cars, and artificial intelligence, can be used to process extensive historical Big Data to solve data-driven problems [28], [29]. Machine learning approaches search for relations between input and output data. After being trained with the training dataset, well-trained machine learning models can be used to make predictions based on the input data. Machine learning algorithms can identify structures and trends in energy datasets, build mathematical models to explain these relationships, and use them to make accurate and reliable predictions about the future generation renewable energy. Model generalisation and feature extraction are two areas where machine learning outperforms traditional statistical predictive models [30].

Machine learning can help make smarter, faster, data-driven estimates about how electricity generation can meet electricity demand [31]. ML can be used for a variety of energy-related tasks, including energy theft detection, demand forecasting, demand side management, optimised energy storage operation, energy generation forecasting, energy price forecasting, predictive maintenance and control, and prediction of weather phenomena that could impact energy forecasting and building energy management. All forms of renewable energy, including wind, solar, geothermal, hydro, marine, bio, hydrogen, and hybrid, can be harnessed with AI models [32].

We have tried our best to improve the manuscript and have made recommended changes in the manuscript. These changes do not affect the scope of the paper. The specific position of the changes in the revised article is explained in detail in this blue-letter response document.

We sincerely thank the editor and reviewers for their work and hope that the corrections will meet with approval.

Once again, thank you very much for your comments and suggestions.